# Metabolic rate evolves rapidly and in parallel with the pace of life history

Sonya K. Auer[1], Cynthia A. Dick[2], Neil B. Metcalfe[1] & David N. Reznick[2]

Metabolic rates and life history strategies are both thought to set the "pace of life", but whether they evolve in tandem is not well understood. Here, using a common garden experiment that compares replicate paired populations, we show that Trinidadian guppy (*Poecilia reticulata*) populations that evolved a fast-paced life history in high-predation environments have consistently higher metabolic rates than guppies that evolved a slow-paced life history in low-predation environments. Furthermore, by transplanting guppies from high- to low-predation environments, we show that metabolic rate evolves in parallel with the pace of life history, at a rapid rate, and in the same direction as found for naturally occurring populations. Together, these multiple lines of inference provide evidence for a tight evolutionary coupling between metabolism and the pace of life history.

[1] Institute of Biodiversity, Animal Health and Comparative Medicine, University of Glasgow, Glasgow G12 8QQ, UK. [2] Department of Biology, University of California-Riverside, Riverside, CA 92521, USA. Correspondence and requests for materials should be addressed to S.K.A. (email: sonya.auer@gmail.com)

Life history traits differ markedly among populations and species[1,2], but tend to fall together along a "slow–fast pace of life continuum", even after accounting for differences in body size. At the slower end of this continuum are organisms that mature at a later age and larger size, reproduce at slower rates, and have longer lifespans; organisms that mature early and at a smaller size, reproduce at a rapid rate, but die young are at the faster end of the continuum[3–7]. Metabolic rate reflects the energetic cost of living and is also thought to set the pace of life[8–10], but whether it evolves alongside the life history is not well understood. Phenotypic correlations between metabolic rate and the pace of life history can be positive, nonsignificant, or negative across populations and species[11–15]. Contrasts between species that differ qualitatively in their general life history, i.e., slow vs fast, also produce mixed results[16,17]. However, comparative studies thus far typically compare populations and species under local rather than common garden environmental conditions. Given that metabolic rates and life history traits are both sensitive to environmental conditions[18,19], the extent to which these patterns of covariation represent purely phenotypic vs genetic (evolutionary) changes is presently unclear.

Here we use a combination of field-transplant experiments and population comparisons under common garden conditions to examine whether and how metabolic rate evolves alongside the life history in Trinidadian guppies (*Poecilia reticulata*). Guppy populations have diverged in the pace of their life history under different predation regimes in their native freshwater streams: populations with a low risk of predation mature at a larger size and older age, reproduce at slower rates, and invest less in reproduction relative to populations from high-predation sites[20–23]. The evolution from a faster- to a slower-paced life history as guppies invaded upstream low-predation reaches has occurred independently within multiple different drainages across Trinidad (see refs. [20,21] and Supplementary Note 1) and can be quite rapid: guppies experimentally transplanted from high- to low-predation sites repeatedly evolve a slower-paced life history in <10 years (~20 generations)[24–26].

Under common garden conditions, we measured standard metabolic rates in laboratory-reared offspring of wild-caught guppies from six populations and three different drainage systems that are known to differ in the pace of their life history[20–22]. Standard metabolic rate reflects the baseline cost of maintaining the tissues and homeostatic mechanisms critical to life[27] and is indicative of total energy expenditure[28,29]. We first compared pairs of populations with slow- vs fast-paced life histories from both the Oropuche and Yarra River drainages, testing for consistent differences in standard metabolic rate across independent evolutionary transitions from a faster- to a slower-paced life history as guppies invaded low-predation environments within different drainages across Trinidad[20,23]. We then evaluated the direction and speed at which standard metabolic rate evolves by comparing its values between a naturally occurring population with a fast-paced life history and a descendant population that was transplanted to a low-predation site in the Caroni River drainage 35 years ago and has since evolved a slower-paced life history[24,25]. Finally, we examined the relationship between standard metabolic rate and the overall pace of life history (male age and mass at maturity, female age and mass at first parturition, reproductive frequency, and reproductive investment) to assess whether patterns of covariation across drainages mirror those found within drainages.

We found that the evolution of a slower-paced life history as guppies invaded upstream low-predation habitats is accompanied by a decrease in standard metabolic rate within each of the three drainages. Furthermore, there is a strong positive association between the rate of metabolism and a suite of life history traits

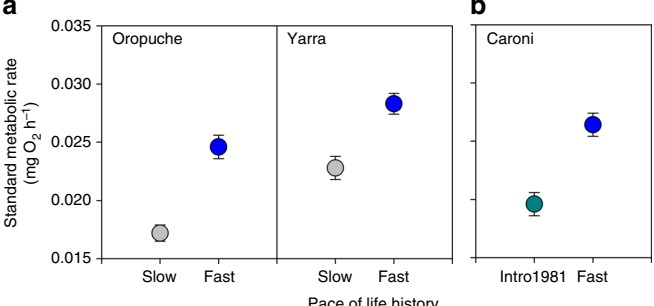

**Fig. 1** Metabolic rates differed among populations according to their pace of life history. **a** Standard metabolic rate (mean ± 1 SE) was lower in naturally occurring populations with a slow (gray) relative to a fast-paced (blue) life history (LMM: $P < 0.001$) in both the Oropuche (slow: $n = 29$, fast: $n = 28$) and Yarra (slow: $n = 26$, fast: $n = 28$) River drainages. **b** Standard metabolic rate (mean ± 1 SE) was also lower in a population transplanted from high- to low-predation sites in 1981 that has since evolved a slow-paced life history (green, $n = 22$) vs their naturally occurring ancestral population with a fast-paced life history (blue, $n = 24$) in the Caroni River drainage. Values for standard metabolic rate are standardized to a common body mass of 74 mg (mean across all fish)

across all populations, regardless of drainage. These parallel changes suggest a tight evolutionary association between metabolic rate and the pace of life history.

## Results

**Naturally occurring populations**. We found consistent differences in standard metabolic rate among naturally occurring guppy populations that exhibit slow- vs fast-paced life histories (Fig. 1a). Standard metabolic rate differed between the Oropuche and Yarra drainages (linear-mixed model (LMM): $F_{1,90.3} = 33.53$, $P < 0.001$) after controlling for effects of body mass (LMM: $F_{1,79.3} = 84.12$, $P < 0.001$); guppies from the Oropuche drainage exhibited a lower standard metabolic rate than those from the Yarra drainage (Fig. 1a). However, standard metabolic rate in the populations with a slow-paced life history was consistently lower than their counterparts with a fast-paced life history across both drainages (Fig. 1a; LMM: $F_{1,92.0} = 55.85$, $P < 0.001$). There was no difference in standard metabolic rate between males and females (LMM: $F_{1,86.4} = 0.36$, $P = 0.551$) or any interactive effects of life history, drainage, and sex (LMM: all $P > 0.05$).

**Experimental evolution**. Additionally, standard metabolic rate evolved rapidly in the experimental population in the Caroni drainage and in the same direction as found above (Fig. 1b). After controlling for effects of body mass (LMM: $F_{1,40.4} = 34.04$, $P < 0.001$), standard metabolic rate diverged between the naturally occurring population with a fast-paced life history and its descendants that have evolved a slow-paced life history since being transplanted into an upstream low-predation site within that same drainage (Fig. 1b). Specifically, the standard metabolic rates of guppies in the introduced population were significantly lower than that of guppies of the same size from the ancestral population with a fast-paced life history (LMM: $F_{1,31.3} = 12.30$, $P = 0.001$). Assuming 1.74 generations per year[25], this minimum rate of evolutionary divergence from the ancestral high-predation population is equivalent to 0.02 haldanes over 60.9 generations, a value that is comparable to those reported for guppy life history traits in the same river, but faster than the average reported thus far for animal taxa in general (Fig. 2)[30]. There was no difference in standard metabolic rate between males and females (LMM:

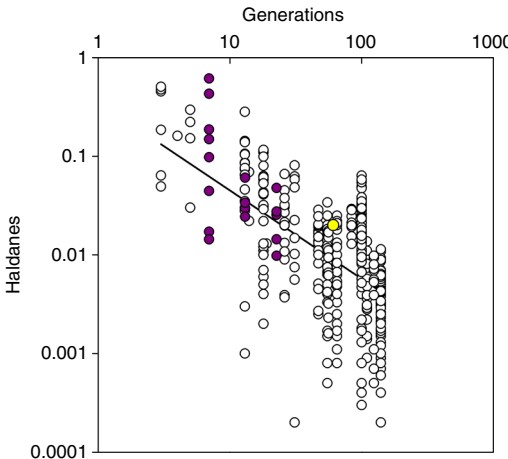

**Fig. 2** Rates of evolution are slower when measured over longer time intervals. Rates, plotted in haldanes (standard deviations of change per generation), are shown for metabolic rate in guppies from the El Cedro River from this study (yellow, $n = 1$), previously published estimates for guppy life history traits from the El Cedro River (purple, $n = 18$)[30], and previously published estimates for life history and morphological traits of other taxa (white, $n = 402$)[30]. All estimates are absolute values of haldanes derived from common garden or quantitative-genetic methods. Note the logarithmic scale on both axes

$F_{1,31.3} = 0.80$, $P = 0.377$) or any interactive effects between sex and life history (LMM: $P > 0.05$).

**Covariation with the pace of life history**. Finally, we found that a single linear function describes the relationship between standard metabolic rate and the pace of life history across all populations, regardless of predation level or drainage. Mean mass-independent standard metabolic rate was positively correlated with the overall pace of life history across the six study populations (Fig. 3; Pearson correlation: $r = 0.93$, $P = 0.007$).

## Discussion

Guppies exhibit consistent changes in their life history when invading low-predation environments[20–22]. Here we consider the role that energy metabolism may play in these evolutionary transitions from a faster- to a slower-paced life history. Our common garden approach comparing naturally occurring and experimentally transplanted populations provide multiple lines of evidence that standard metabolic rate evolves together and in a positive direction with the pace of life history. This relationship is sustained across genetically divergent populations[31], indicating parallelism among different lineages. Consistent differences in metabolic rate and life history traits between low- and high-predation populations under common garden conditions suggest genetically based divergence in the wild[32]. Maternal effects may have also contributed to observed differences but are unlikely, given that they are generally found to be absent or negligible in studies of metabolic rate, see e.g., refs [33–37]. In addition, we observed an effect of drainage that was independent of predation (Fig. 1), suggesting that evolutionary history may also play a role in how these traits coevolve (see Supplementary Note 1).

These parallel evolutionary changes suggest a tight association between metabolic rate and the life history. Indeed, our study and others show that metabolic rate[38–44] and the life history[25,45–47] can evolve in response to some of the same environmental factors —including predation, disease, climate, food availability, diet quality, and population density. While it is not yet clear whether metabolic and life history traits are responding to the same or

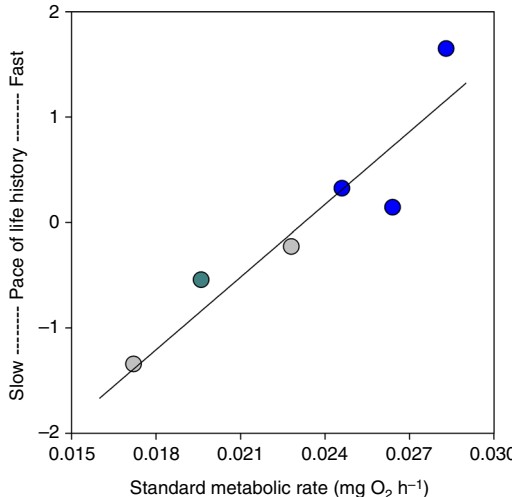

**Fig. 3** Standard metabolic rate covaries with the pace of life history. Standard metabolic rate was positively correlated with a suite of life history traits across six populations (Pearson correlation: $P = 0.007$). Included are naturally occurring populations with fast-paced ($n = 3$) and slow-paced life histories (gray, $n = 2$) and a population transplanted from a high- to low-predation site 35 years ago that has since evolved a slow-paced life history (green, $n = 1$). The pace of life history was determined using principal component analysis (PCA) of estimates for the suite of traits including male age and mass at maturity, female age and mass at first parturition, inter-litter interval, and reproductive allotment. The first principle component (plotted on the $y$ axis) had an eigenvalue of 3.70, explained 61% of the variation in life history traits, and differentiated populations with later maturity and lower reproductive investment (lower scores) from those with earlier maturity and higher reproductive investment (higher scores). Values for standard metabolic rate were standardized to a common body mass of 74 mg prior to analysis (mean across all fish)

different agents of selection, quantitative-genetic and physiological studies suggest a functional linkage between them that could cause them to evolve together. In particular, there is some evidence for a positive genetic correlation between metabolic rate and the pace of life history[48–50]. Species with a higher standard metabolic rate also tend to have higher levels of activity and total energy expenditure[28,29]. Faster metabolic rates may therefore enable faster rates of energy acquisition that then promote faster rates of somatic growth and reproduction[51–53]. However, they can also lead to increased predation risk and/or cause biological damage that is costly to survival[54–57]. As such, trade-offs in the allocation of metabolic power among competing functions may limit the combination of possible trait values and provide a unifying mechanistic explanation for why life history strategies across a broad array of taxonomic groups tend to evolve along a slow–fast continuum[9].

## Methods

**Ethics statement**. Experiments were conducted in accordance with approved protocols from the Institutional Animal Care and Use Committee of the University of California-Riverside (Animal Use Protocol No: A-20170006).

**Source populations**. Wild pregnant females (10–15 fish) were collected in the summer of 2016 from populations inhabiting three separate river systems draining the Northern Range Mountains of Trinidad, West Indies. Parental fish were collected from populations in low-predation (LP) and high-predation (HP) sites within the northern slope Yarra drainage (Yarra-Limon LP and Yarra 3 HP sites[21]) and within the southeastern slope Oropuche drainage (Quare 6 LP and Oropuche 2 HP sites[22]). Guppies were also collected from two sites in the southwestern slope Caroni drainage: a high-predation site (El Cedro 1 HP site[22]) and an upstream low-predation site where fish from that same high-predation site were introduced in

1981 (fish taken from El Cedro 1 HP site[18] and moved to El Cedro tributary[25,26]). These populations were selected because guppies from the LP and HP sites within each of the three drainages are known from previous laboratory common garden studies to exhibit slow- vs fast-paced life histories, respectively[20,23].

**Rearing of F1 experimental fish**. Wild-caught guppies were transported to the University of California-Riverside, where each population was housed separately in 10-gallon tanks. Females can produce multiple broods using stores of viable sperm from multiple males[58], so they continued to produce offspring even in the absence of males. The tanks were checked daily, and new-born F1 offspring from each population were transferred and housed together in 5-gallon tanks. F1 offspring were born under communal conditions, so we were not able to quantify the number of offspring produced by each female that were included in the experiment. However, females reproduce at regular intervals, so experimental fish represent a random sample of the F1 offspring produced by all wild-caught mothers.

Once experimental (F1) fish began to show signs of maturity, males from each population were transferred to a separate 5-gallon tank (11−16 fish per population per tank). Maturing females were distinguished from males by the triangular pattern of melanophore development in their abdominal region, while males were identified by their partially (immature) or fully developed (mature) intromittent organ[59]. Both parent and F1 tanks were maintained at $25.0 \pm 0.5 °C$ (mean ± actual range) and under a 12D:12L cycle. The fish were fed to satiation twice daily with liver paste in the morning and *Artemia* brine shrimp nauplii in the afternoon.

Transgenerational effects are a common concern when inferring genetic from phenotypic differences between populations or species under common garden conditions. However, an equally valid concern is that of inadvertent selection for particular genotypes favored by artificial laboratory conditions[32,60]. As such, we chose to focus on the F1 generation because it is an optimal resolution[32] to the trade-off between minimizing potential environmental/maternal effects while also minimizing the erosion of genetic diversity that can occur due to (unintentional) selection in the laboratory environment. In addition to our use of common garden conditions, our experimental protocol also minimized potential effects of field conditions (via maternal or environmental effects) by including in the experiment only those F1 offspring yolked under these common laboratory conditions. This was verified by using known growth trajectories[61] and inter-litter intervals (21.6−24.7 days)[23] of guppies under similar laboratory conditions to back-calculate to the time experimental fish were yolked and conceived. Based on their body size at the time of the experiment, all experimental fish were yolked and conceived 2 or more months after the time their mothers first arrived in the laboratory.

**Metabolic rate measurements**. Standard metabolic rate was measured in juvenile and adult male and female F1 guppies (Supplementary Table 1). Metabolic rates were measured using continuous flow through respirometry in the same temperature-controlled laboratory during the autumn of 2016 and using the methodology and instrumental set-up described in ref. [62]. Briefly, water in the respirometry system was run through a UV sterilizer (Aqua Ultraviolet, Temecula CA) to minimize background respiration rates. Water was pumped from an aerated upper bin through individual oxygen impermeable Tygon tubes (Cole Parmer, London, UK) to each of 16 respirometry chambers (volume 20 ml) arranged in parallel and submerged in a lower water bath, then via additional tubing past an oxygen sensor (robust probe; PyroScience GmbH, Aachen, Germany) sealed inside a small chamber before draining into a lower bin and being recirculated back up to the upper bin. A peristaltic pump (Cole Parmer, London, UK) pulled water through the system at a constant rate and was used to adjust the flow rate through each chamber.

Fish were isolated from their stock tanks and fasted for 24 h before their metabolic measurements to ensure they were in a postabsorptive state. Dissection analysis indicated that this time interval is sufficient for them to fully evacuate their guts and thereby prevent digestive costs from biasing measurements of standard metabolism. Oxygen sensors were calibrated using fully aerated water before each batch of fish was put in the chambers. Fish were then placed in their respirometry chambers in the early afternoon, and their oxygen consumption was measured continuously over the next 20 h. The water bath containing the respirometry chambers was covered with a sheet of black plastic on all sides to keep fish activity to a minimum. The flow rate was set to 0.84 ml/min as this allowed us to detect their oxygen consumption while ensuring that oxygen levels in the chambers never dipped below 80% saturation. Oxygen levels were measured every 2 s by four multichannel oxygen meters and associated software (FireStingO2, PyroScience).

Oxygen consumption was generally measured in one male and one female per population per day (a total of 14 batches over 15 days), except for the last two batches where we included mostly females to increase sample sizes of females differing in their reproductive status (see below). The remaining two fish-free chambers served as a control of background respiration. Standard metabolic rate (mg $O_2$ h$^{-1}$) was measured as: $M_{O2} = V_w \times (C_{wO2control} - C_{wO2fish})$, where $V_w$ is the flow rate of water through the respirometry chamber (L h$^{-1}$), and $C_{wO2control}$ and $C_{wO2fish}$ are the concentrations of oxygen (mg L$^{-1}$) in the outflow of the chambers lacking and containing fish, respectively, after adjusting for temperature and barometric pressure[63]. Standard metabolic rate for each fish was calculated by taking the mean of the lowest 10$^{th}$ percentile of oxygen consumption

measurements over the 20 h measurement period, and then excluding outliers, i.e., those measurements below 2 standard deviations from this mean[63]. Several fish did not settle down in their respirometry chambers, so they were excluded from further analyses (see Supplementary Table 1 for final sample sizes). After measurement of metabolic rates, fish were removed from the system, euthanized with a fatal overdose of neutrally buffered MS-222 (Sigma-Aldrich, St. Louis, MO), weighed (±0.001 g), and preserved in 5% formalin for later dissection analysis. The respirometry system was then cleaned using bleach and UV-sterilized tap water before the next batch of measurements.

**Determination of female reproductive status**. Females were maintained in their own sex-specific stock tanks, but could still yolk eggs in the absence of males. Female guppies are lecithotrophic, meaning that they provide no additional resources to their young after yolking their eggs[64]. However, the effect of pregnancy status on standard metabolic rate is not known. As such, females were dissected under a microscope and their reproductive tissues were removed to assess their stage of reproduction. Females were then weighed again to determine their non-reproductive wet mass. Most females had no eggs (54%) or undeveloped eggs (38%), but some females (8%) carried developing embryos because the maturation of some males went initially undetected, while they were housed together in communal stock tanks. Five females were in the process of yolking their eggs, so they were excluded from analyses.

**Characterization of the pace of life history**. The pace of life history was characterized using data on the entire suite of guppy life history traits including male age and mass at maturity, female age and mass at first parturition, inter-litter interval, and reproductive allotment (Supplementary Table 2). Life history traits were measured in separate laboratory studies that compared the F2 generation of populations from low- vs high-predation sites within each of the three drainages[20,23]. Life history traits for the naturally occurring high-predation population and its descendant population in the Caroni River drainage were evaluated 13 years post introduction. All three laboratory studies were conducted under similar environmental conditions with respect to light (12D:12L) and diet (liver paste in the morning and *Artemia* brine shrimp in the afternoon as done here). One of the two food levels used in each study (low food levels from[20] and high food level from[23]) sustained roughly 60–70% of maximum growth, so food levels were also comparable across all three studies. All three studies used the same protocols with respect to the rearing of laboratory stocks and the measurement of life history traits in fish. F2 males and females from each population were reared individually starting from age 21 to 25 days until they matured, in the case of males, or had given birth to their third litter, in the case of females. Morphogenesis of the anal fin was used to determine male age and size at maturation; specifically, males were considered mature if their 3rd anal fin ray had fully developed into their intromittent organ, the gonopodium[59]. Female age and size at first parturition were determined at the birth of their first litter. Inter-litter interval is a measure of reproductive frequency and was calculated as the mean time interval (in days) between the births of the first and second and between the second and third litters. Reproductive allotment was calculated as % female dry weight devoted to offspring in the third litter and used as measure of reproductive effort that takes into account the positive effects of female size on reproductive output.

**Statistical analyses**. We first assessed potential differences in standard metabolic rate among paired guppy populations with slow- vs fast-paced life histories from the Yarra and Oropuche drainages. The general linear model included the pace of life history (slow vs fast), drainage (Oropuche vs Yarra), and their interaction as fixed categorical factors, body mass as a covariate, and respirometry batch and chamber number as random effects. Both standard metabolic rate and body mass were log$_{10}$-transformed to linearize the data. The slope of metabolic rate with body mass did not differ between the four populations (population by log$_{10}$-transformed body mass: $P = 0.496$).

We then evaluated the direction and rate at which standard metabolic rate evolves by comparing its values between a naturally occurring population with a fast-paced life history and its descendants that were transplanted to a low-predation site in the Caroni River drainage 35 years ago and have since evolved a slower-paced life history. Both standard metabolic rate and body mass were log$_{10}$-transformed to linearize the data. The model included pace of life history (slow vs fast) as a fixed categorical effect, body mass as a covariate, and respirometry batch and chamber number as random effects. The slope of metabolic rate with body mass did not differ between the two populations (population by log$_{10}$-transformed body mass: $P = 0.389$).

Reproductive status (no eggs vs eggs or embryos) did not affect the standard metabolic rate of females (Supplementary Table 3), so males and females were analyzed together and sex was included as a fixed categorical effect in the same models. Growth may also contribute to metabolic costs. However, if growth influenced SMR, then we would expect to see differences in metabolic rate between those individuals that are still growing (juveniles and females) and those that have matured and stopped growing (males). Instead, we find no effect of life stage ($P = 0.367$) or sex (see Results). As such, immature and mature individuals were analyzed together.

Evolutionary rates of divergence were quantified using haldanes. Haldanes measure the change in a phenotypic trait, in units of standard deviations, over time and were calculated as follows[65,66]: $[(\ln x_2/s_{\ln x})-(\ln x_1/s_{\ln x})]/t$ where $\ln x_1$ and $\ln x_2$ are sample means of natural log-transformed measurements and the standard metabolic rate in the introduced and ancestral population, respectively, $t$ is the time interval (in number of generations) over which evolution was measured, and $s_{\ln x}$ is the pooled standard deviation of $\ln x_1$ and $\ln x_2$. Rates for guppies were calculated assuming 1.74 generations per year[25]. The absolute values of evolutionary rates decline as a logarithmic function of the interval over which they are measured[65,66], so the observed rate of divergence in standard metabolic rate was $\log_{10}$-transformed and plotted against $\log_{10}$-transformed time interval (in generations) in order to compare it with published estimates from previous studies of the evolution of life history traits in guppies in that same population and life history and morphological traits in other animal taxa.

Finally, we used correlation analysis (Pearson's $r$) to assess the covariation between standard metabolic rate and the overall pace of life history. Residuals from the model relating $\log_{10}$-transformed standard metabolic rate (SMR) to $\log_{10}$-transformed body mass (BM; $\log_{10}$-SMR $= -0.95 + 0.61*\log_{10}$-BM) were used to standardized standard metabolic rate to a common body mass of 74 mg (mean across all fish). The pace of life history was determined using principal component analysis (PCA) of population level estimates for the suite of traits that define the life history including male age and mass at maturation, female age and mass at first parturition, inter-litter interval, and reproductive allotment (Supplementary Table 2). The first PCA had an eigenvalue of 3.70 and described 61% of the variation in life history traits across populations. Loadings along PCA 1 characterized the slow-to-fast life history continuum across guppy populations: -0.763 for male size at maturity, -0.866 for male age at maturity, -0.843 for female age at first parturition, -0.744 for female size at first parturition, -0.677 for inter-litter interval, and 0.806 for reproductive allotment. PCA thus arrayed the populations from ones with rapid development (early maturity) and high investment in reproduction to those with slow development and low investment in reproduction. This analysis was conducted on the mean standard metabolic rate and the single PCA score for each population.

Data were analyzed using SPSS version 22.0 (IBM statistics). Fresh body mass was highly correlated with formalin-preserved body mass in males and non-reproductive females (Pearson correlation: $r = 0.99$, $P < 0.001$, $n = 124$), and the slope of their relationship did not differ from unity (slope $\pm$ SE: $1.07 \pm 0.02$). As such, we used measurements of formalin-preserved body mass in all our analyses of metabolic rate since this allowed us to include reproductive females and their non-reproductive body masses determined after preservation and dissection. Spontaneous activity of fish in the respirometry chambers was minimal, but may still bias estimates of standard metabolic rate. However, we found no evidence for such effects: there was no correlation between mass-independent standard metabolic rate and mass-independent excess routine metabolic rate (the excess oxygen expended on spontaneous activity above standard metabolic rate; Pearson correlation: $r = -0.09$, $P = 0.225$, $n = 157$).

**Data availability**. Data supporting the findings of this study are available in the manuscript, its supplementary files, and from the corresponding author upon request.

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

## Acknowledgements

The authors thank Joshua Goldberg and Robert Prather for collecting fish in Trinidad and Yuridia Reynoso for helping to maintain the laboratory stocks. This research was funded by a University of Glasgow Lister Bellahouston Travelling Fellowship to S.K.A., a US National Science Foundation Pre-Doctoral Fellowship to C.A.D., a European Research Council Advanced grant (no. 322784) to N.B.M., and US National Science Foundation grants (DEB-0623632EF and DEB-1258231) to D.N.R.

## Author contributions

S.K.A. conceived the study with input on experimental protocol from N.B.M. and D.N.R. C.A.D. and D.N.R reared and maintained fish under standardized conditions in the laboratory. S.K.A. and C.A.D. undertook the experiment. S.K.A. analyzed the data and drafted the manuscript. All authors contributed to subsequent revisions.

## Additional information

**Competing interests:** The authors declare no competing financial interests.

