## [Peer Review File · Nature Communications]

Reviewers' comments:

Reviewer #1 (Remarks to the Author):

I was very excited to read this manuscript, which presents the results of a topical study of a potentially powerful study system. The manuscript is well written and well presented, and in general the study is extremely well conducted. However, as it stands, I have one potentially very significant concern. Although few studies have been conducted, there is evidence of maternal contributions to phenotypic variance in metabolic rate for a range of species (e.g. Nespolo et al. 2003; Régnier et al. 2010). Given that you measured laboratory-reared offspring (F1) of wild-caught guppies, how can you be sure that the phenotypic differences you observe among animals from different predation regimes do not arise as a consequence of transgenerational effects of the maternal environment rather than genetic differences between populations? Without measurement of F2 animals (the laboratory-reared offspring of laboratory-reared adults), how are you able to be sure that the differences you observe are genetic?

Minor comments

L28-33. Given that you point out that the life-history continuum is present independent of body mass, I think that it would be preferable to remove the references to body size in the second sentence of this paragraph (the one beginning "At the slower end..." Alternatively, if you mean that animals at the slower end mature at a larger fraction of maximum body size, then it might be best to clarify this.

L57. In fish that show indeterminate growth, SMR might also include a contribution of growth (Rosenfeld et al. 2015), and so may not represent a genuine measure of maintenance costs alone unless animals maintained on a maintenance ration.

Nespolo, R. F., L. D. Bacigalupe, and F. Bozinovic. 2003. Heritability of energetics in a wild mammal, the leaf-eared mouse (*Phyllotis darwini*). *Evolution* 57:1679-1688.

Régnier, T., V. Bolliet, J. Labonne, and P. Gaudin. 2010. Assessing maternal effects on metabolic rate dynamics along early development in brown trout (*Salmo trutta*): an individual-based approach. *J. Comp. Physiol. B* 180:25-31.

Rosenfeld, J., T. Van Leeuwen, J. Richards, and D. Allen. 2015. Relationship between growth and standard metabolic rate: measurement artefacts and implications for habitat use and life-history adaptation in salmonids. *J. Anim. Ecol.* 84:4-20.

Reviewer #2 (Remarks to the Author):

Auer et al present an observational/experimental study investigating population differences in Standard Metabolic Rate (SMR) and 'pace of life' as measured by a PCA of standard life

history traits in Trinidad guppies. They find a positive correlation between SMR and pace of life and show that both traits evolve in the same direction after experimental introductions from high to low predation. Overall, this is a really nice study that reveals an interesting pattern. The ms is very clear and well written. However, the broader implications are somewhat underdeveloped. While I appreciate a concise discussion, I found the two paragraphs of discussion to be somewhat limiting. The first is fairly guppy-centric, and perhaps overplays the importance of genetic correlations, which the authors have not measured. The second makes the point that evolutionary and plastic responses may be important in population responses to climate change, but again, there's not a lot of context here.

In the introduction, the authors state that other studies have examined the same two traits and variously found positive, negative, or no correlation. This study finds a positive correlation. Why do the authors think it is positive in this case and not others? If I'm reading correctly, this is the only study to use common garden experiments in this context, yet their results show that they would have gotten the same results comparing field-collected samples. So, is there something about the system that predicts this positive relationship, and something about other systems that predicts negative or no correlations? More discussion here would be helpful. As is, it is difficult to assess the overall novelty and impact of this paper without better framing.

Though the F1 common garden approach is a real strength, the authors ought to at least acknowledge the possibilities of maternal effects, or rule them out based on previous research.

Finally, though the F statistics include degrees of freedom, it's tough to judge the true level of replication in the study. The lab portion uses offspring produced from field-caught females, but the authors do not say how many different females produced the experimental animals, i.g., how many sib families are represented in each treatment combination?

Other comments:

L33: 'energetic cost of living' probably bears a little more explanation given that it is a central point in the paper.

Fig 1. Is there a reason why body mass can't just be added into the units, i.e., mgO₂/gh. That's a pretty standard unit that accounts for body mass. Also, colors aren't really important because everything is in separate panels, but if you are going to use them, the yellow seems to have printed a little funny, and is difficult to distinguish from the green.

Fig 2. Why don't the colors align with Fig 1?

L115. Could be a genetic association, but given the evidence presented it could just as easily be driven by correlational selection.

L244. The intro/discussion doesn't appear to talk much about evolutionary rates, except to

say that evolution is 'rapid'. Perhaps using a metric such as a Haldane would help to put 'rapid' in perspective.

Reviewer #3 (Remarks to the Author):

Authors use a combination of experimental approaches to convincingly demonstrate that a presumed proxy of pace-of-life (size at maturity) co-evolves with metabolic rates in guppies. The paper is timely, well written, the experimental design sound, the results properly analyzed, and the interpretation straightforward.

My only major comment is that pace of life is defined here as size at maturity, which is convenient (because easily measured) but not necessarily appropriate. Variation in pace of life is typically viewed as variation among species, populations, or individuals in how the trade-off between current and future reproduction is resolved. Useful metrics of pace-of-life include age at first reproduction, average age of reproduction ('generation time'), longevity, or a multivariate summary of these metrics.

As the paper essentially does not measure aspects of life-history apart from a morphological trait (adult size), it is hard to place it in the current hot literature on pace-of-life evolution. Authors should therefore either include evidence of covariance with more convincing metrics of pace of life into their study or reframe the rationale of this study entirely.

Reviewers' comments:

Reviewer #1 (Remarks to the Author):

Comment: I was very excited to read this manuscript, which presents the results of a topical study of a potentially powerful study system. The manuscript is well written and well presented, and in general the study is extremely well conducted. However, as it stands, I have one potentially very significant concern. Although few studies have been conducted, there is evidence of maternal contributions to phenotypic variance in metabolic rate for a range of species (e.g. Nespolo et al. 2003; Régnier et al. 2010). Given that you measured laboratory-reared offspring (F1) of wild-caught guppies, how can you be sure that the phenotypic differences you observe among animals from different predation regimes do not arise as a consequence of transgenerational effects of the maternal environment rather than genetic differences between populations? Without measurement of F2 animals (the laboratory-reared offspring of laboratory-reared adults), how are you able to be sure that the differences you observe are genetic?

Nespolo, R. F., L. D. Bacigalupe, and F. Bozinovic. 2003. Heritability of energetics in a wild mammal, the leaf-eared mouse (*Phyllotis darwini*). *Evolution* 57:1679-1688.

Régnier, T., V. Bolliet, J. Labonne, and P. Gaudin. 2010. Assessing maternal effects on metabolic rate dynamics along early development in brown trout (*Salmo trutta*): an individual-based approach. *J. Comp. Physiol. B* 180:25-31.

Reply: Transgenerational effects are a common concern when inferring genetic from phenotypic differences between populations or species under common garden conditions. However, an equally valid concern is that of inadvertent selection for particular genotypes favoured by artificial laboratory conditions (Garland and Adolph 1991; Christie et al 2012). As such, we chose to focus on the F1 generation because it is an optimal resolution (Garland and Adolph 1991) to the trade-off between minimizing potential environmental/maternal effects while also minimizing the erosion of genetic diversity that can occur due to (unintentional) selection in the laboratory environment. We acknowledge the potential for maternal effects in the F1 generation but think it unlikely that they influenced population differences in life history and metabolic rate for several reasons.

First, population differences in life history observed among F1 fish in this study mirror those found for F2s of these same populations and drainages under common garden conditions (Reznick et al 2004), suggesting that maternal effects, if present, contribute little to observed variation in life history traits. In addition, there is currently little evidence for maternal effects on metabolic rates. Of those studies that are able to tease apart the relative effects of environmental versus maternal versus dominant genetic effects on observed phenotypic variation, maternal effects were reported as absent or negligible (see Sadowska et al 2005, Wone et al 2009, Nespolo et al 2005, Ronning et al 2007, Bacigalupe et al 2004, Zub et al 2012, but see also Nespolo et al 2003 cited by reviewer above). In the other reference cited above by the reviewer (Regnier et al 2010), female identity explained only 1% of the total variance in metabolic rate and it is not clear if this effect represented a genetic or nongenetic maternal effect.

There is some evidence that exposure to experimentally administered hormones such as cortisol can affect offspring metabolic rates (e.g. Sloman 2010, Nilsson et al 2011), so maternal/environmental effects have the potential to contribute at least in part to some of the observed phenotypic variation. For this reason, our experimental protocol minimized potential effects of field conditions (via maternal or environmental effects) in two ways: First, F1 fish from all populations were raised under common environmental conditions. Second, only those offspring yolked under common laboratory conditions were used in the experiment. This was verified by using known growth trajectories (Auer 2010) and inter-litter intervals (21.6-24.7 days; Reznick 1982) of guppies under similar laboratory conditions to back-calculate to the time experimental fish were yolked and conceived. Based on their body size at the time of the experiment, all experimental fish were yolked and conceived more than two months after the time their mothers first arrived in the laboratory. The latter was not properly explained in the methods, so we have now added these details (see lines 173-179).

- Auer, S. K. Phenotypic plasticity in adult life history strategies compensates for a poor start in life in Trinidadian Guppies (*Poecilia reticulata*) *Am. Nat.* 176, 818-829 (2010).
- Bacigalupe, L. D., Nespolo, R. F., Bustamante, D. M., Bozinovic, F. & Sinervo, B. The quantitative genetics of sustained energy budget in a wild mouse. *Evolution* 58, 421-429 (2004).
- Christie, M. R., Marine, M. L., French, R. A. & Blouin, M. S. Genetic adaptation to captivity can occur in a single generation. *Proc. Natl. Acad. Sci. U. S. A.* 109, 238-242 (2012).
- Garland, T. & Adolph, S. C. Physiological differentiation of vertebrate populations. *Annu. Rev. Ecol. Syst.* 22, 193-228 (1991).
- Nespolo, R. F., Bustamante, D. M., Bacigalupe, L. D. & Bozinovic, F. Quantitative genetics of bioenergetics and growth-related traits in the wild mammal, *Phyllotis darwini*. *Evolution* 59, 1829-1837 (2005).
- Nilsson, J. F., Tobler, M., Nilsson, J.-Å. & Sandell, M. I. Long-lasting consequences of elevated yolk testosterone for metabolism in the zebra finch. *Physiol. Biochem. Zool.* 84, 287-291 (2011).
- Reznick, D. The impact of predation on life history evolution in Trinidadian guppies: Genetic basis of observed life history patterns. *Evolution* 36, 1236-1250 (1982).
- Reznick, D. N., Bryant, M. J., Roff, D., Ghalambor, C. K. & Ghalambor, D. E. Effect of extrinsic mortality on the evolution of senescence in guppies. *Nature* 431, 1095-1099 (2004).
- Rønning, B., Jensen, H., Moe, B. & Bech, C. Basal metabolic rate: heritability and genetic correlations with morphological traits in the zebra finch. *J. Evol. Biol.* 20, 1815-1822 (2007).
- Sadowska, E. T. et al. Genetic correlations between basal and maximum metabolic rates in a wild rodent: consequences for evolution of endothermy. *Evolution* 59, 672-681 (2005).
- Sloman, K. A. Exposure of ova to cortisol pre-fertilisation affects subsequent behaviour and physiology of brown trout. *Horm. Behav.* 58, 433-439 (2010).
- Wone, B. et al. A strong response to selection on mass-independent maximal metabolic rate without a correlated response in basal metabolic rate. *Heredity* 114, 419-427 (2015).

Zub, K., Piertney, S., Szafranska, P. A. & Konarzewski, M. Environmental and genetic influences on body mass and resting metabolic rates (RMR) in a natural population of weasel *Mustela nivalis*. *Mol. Ecol.* **21, 1283-1293 (2012).**

Minor comments

Comment: L28-33. Given that you point out that the life-history continuum is present independent of body mass, I think that it would be preferable to remove the references to body size in the second sentence of this paragraph (the one beginning "At the slower end...". Alternatively, if you mean that animals at the slower end mature at a larger fraction of maximum body size, then it might be best to clarify this.

Reply: The slow-fast life history continuum is determined in large part by body size. However, the continuum is still present after correcting for body size effects. Given the influence of body size (e.g. mouse versus elephant), it is important to acknowledge that the continuum also lies along a second axis that is independent of body size (see Dobson 2007, Sibly and Brown 2007).

Dobson, F. S. A lifestyle view of life-history evolution. *Proc. Natl. Acad. Sci. U. S. A.* **104, 17565-17566 (2007).**

Sibly, R. M. & Brown, J. H. Effects of body size and lifestyle on evolution of mammal life histories. *Proceedings of the National Academy of Sciences of the United States of America* **104, 17707-17712 (2007).**

Comment: L57. In fish that show indeterminate growth, SMR might also include a contribution of growth (Rosenfeld et al. 2015), and so may not represent a genuine measure of maintenance costs alone unless animals maintained on a maintenance ration. Rosenfeld, J., T. Van Leeuwen, J. Richards, and D. Allen. 2015. Relationship between growth and standard metabolic rate: measurement artefacts and implications for habitat use and life-history adaptation in salmonids. *J. Anim. Ecol.* 84:4-20.

Reply: The reviewer is correct that SMR may include the cost of growth. However, if growth influenced SMR, then we would expect to see differences in metabolic rate between those individuals that are still growing (juveniles and females) and those that have matured and stopped growing (males). Instead, we find no effect of life stage ($p = 0.616$) or sex (see statistical analysis section of methods). As such, we are confident that growth did not explain differences in SMR observed between populations.

Reviewer #2 (Remarks to the Author):

Comment: Auer et al present an observational/experimental study investigating population differences in Standard Metabolic Rate (SMR) and 'pace of life' as measured by a PCA of standard life history traits in Trinidad guppies. They find a positive correlation between SMR and pace of life and show that both traits evolve in the same direction after experimental introductions from high to low predation. Overall, this is a really nice study that reveals an interesting pattern. The ms is very clear and well written. However, the broader implications

are somewhat underdeveloped. While I appreciate a concise discussion, I found the two paragraphs of discussion to be somewhat limiting. The first is fairly guppy-centric, and perhaps overplays the importance of genetic correlations, which the authors have not measured. The second makes the point that evolutionary and plastic responses may be important in population responses to climate change, but again, there's not a lot of context here.

Reply: We are constrained by space limitations but have now revised the discussion section to focus less on guppies and more on the broader implications of our results – namely the tight evolutionary coupling between metabolic rate and the life history and the mechanisms underlying their relationship (see lines 115-139).

Comment: In the introduction, the authors state that other studies have examined the same two traits and variously found positive, negative, or no correlation. This study finds a positive correlation. Why do the authors think it is positive in this case and not others? If I'm reading correctly, this is the only study to use common garden experiments in this context, yet their results show that they would have gotten the same results comparing field-collected samples. So, is there something about the system that predicts this positive relationship, and something about other systems that predicts negative or no correlations? More discussion here would be helpful. As is, it is difficult to assess the overall novelty and impact of this paper without better framing.

Reply: The reviewer rightly points to a contradiction between the novelty of our study and our comparisons of laboratory and field estimates. The novelty of our study (in addition to the inclusion of experimental evolution) is that we use a common garden experimental protocol to infer genetic differences among populations and their rates of evolution. Common garden conditions are critical for separating genetic from environmental effects. Population differences under common garden conditions *do not* always mirror those found in the field (e.g. see Reznick et al 1987 for example in guppies), and it is these inconsistencies that likely explain differences in results among studies based solely on field measures. As such, it is inappropriate for us to compare field and laboratory measures in the same analysis. We have therefore omitted our analysis of *a*) metabolic rate in the laboratory versus male size at maturity in the wild (former Fig. 2) and *b*) male size at maturity in the laboratory versus the pace of the life history in the wild (former Supplemental Fig. 1). We have also replaced former Supplemental Fig. 1 and associated analyses with separate analyses and figures of the relationship between male size at maturity and the pace of life under laboratory conditions and under field conditions (see new Suppl. Fig. 1).

Reznick, D. N. & Bryga, H. Life-history evolution in guppies (*Poecilia reticulata*): 1. Phenotypic and genetic changes in an introduction experiment. *Evolution*, 1370-1385 (1987).

Comment: Though the F1 common garden approach is a real strength, the authors ought to at least acknowledge the possibilities of maternal effects, or rule them out based on previous research.

Reply: See response to Reviewer 1 above.

Comment: Finally, though the F statistics include degrees of freedom, it's tough to judge the true level of replication in the study. The lab portion uses offspring produced from field-caught females, but the authors do not say how many different females produced the experimental animals, i.g., how many sib families are represented in each treatment combination?

Reply: The experimental fish were offspring of wild-caught females. They were born in communal stock tanks in the laboratory, so we have no way of knowing how many families are represented in our sample. As such, we were not able to include family as a random effect. However, guppies reproduce at regular intervals, so what we do know is that we took a random sample from the offspring of all field-caught mothers (10-15 mothers per population). We have now explained this in the methods section (lines 157-165).

Other comments:

Comment: L33: 'energetic cost of living' probably bears a little more explanation given that it is a central point in the paper.

Reply: By energetic cost of living we mean rates of substrate oxidation or the amount of energy used per unit time. However, we prefer to refer to it first in a general way that is more intuitive to the reader and then to define it more specifically when describing our study (lines 58-60).

Comment: Fig 1. Is there a reason why body mass can't just be added into the units, i.e., mgO₂/gh. That's a pretty standard unit that accounts for body mass.

Reply: Metabolic rate is calculated on a per gram basis. However, we prefer to continue presenting it this way since labelling it as mg O₂ g⁻¹ h⁻¹ may lead readers to erroneously assume we disregarded the nonlinear relationship between metabolism and body mass and simply divided metabolic rate by body mass (which is commonly done, but which is an incorrect method of correcting for mass). We have now included a more detailed explanation of our derivation of mass-independent metabolic rate in both figure legends (see Figs 1 and 2).

Comment: Also, colors aren't really important because everything is in separate panels, but if you are going to use them, the yellow seems to have printed a little funny, and is difficult to distinguish from the green. Fig 2. Why don't the colors align with Fig 1?

Reply: Good point. We have aligned the colours in all figures, including the supplementary material.

Comment: L115. Could be a genetic association, but given the evidence presented it could just as easily be driven by correlational selection.

Reply: We agree with the reviewer. The revised discussion includes a statement to this effect, i.e. that metabolic rate and life history could be products of the same cause (underlying genes) or independent, correlated causes (different genes)(see lines 124-139).

Comment: L244. The intro/discussion doesn't appear to talk much about evolutionary rates, except to say that evolution is 'rapid'. Perhaps using a metric such as a Haldane would help to put 'rapid' in perspective.

Reply: The reviewer makes a good point here. "Rapid" needs to be qualified. We have now calculated evolutionary rates of divergence (in haldanes) between the ancestral and introduced populations. Because evolutionary rates decline as a logarithmic function of the time interval (in generations) over which they are evaluated, we also compared rates in haldanes that we found with those reported in previous studies of guppies and other taxa (see results lines 92-95, lines 103-106, and new Fig. 3).

Reviewer #3 (Remarks to the Author):

Authors use a combination of experimental approaches to convincingly demonstrate that a presumed proxy of pace-of-life (size at maturity) co-evolves with metabolic rates in guppies. The paper is timely, well written, the experimental design sound, the results properly analyzed, and the interpretation straightforward.

Comment: My only major comment is that pace of life is defined here as size at maturity, which is convenient (because easily measured) but not necessarily appropriate. Variation in pace of life is typically viewed as variation among species, populations, or individuals in how the trade-off between current and future reproduction is resolved. Useful metrics of pace-of-life include age at first reproduction, average age of reproduction ('generation time'), longevity, or a multivariate summary of these metrics. As the paper essentially does not measure aspects of life-history apart from a morphological trait (adult size), it is hard to place it in the current hot literature on pace-of-life evolution. Authors should therefore either include evidence of covariance with more convincing metrics of pace of life into their study or reframe the rationale of this study entirely.

Reply: Thank you for bringing this to our attention. We have now provided additional justification for using male size at maturity as an index of the pace of the life history. Specifically, male size at maturity is genetically and phenotypically positively correlated with age at maturity in guppies (Reznick et al 1997). This information was not included in the manuscript but adds additional rationale for why male size at maturity is a good index of the pace of the life history in guppies, as it is in other species. We have now explained this in the introduction (see lines 56-58).

Reznick, D. N., Shaw, F. H., Rodd, F. H. & Shaw, R. G. Evaluation of the rate of evolution in natural populations of guppies (*Poecilia reticulata*). *Science* 275, 1934-1937 (1997).

Reviewers' comments:

Reviewer #1 (Remarks to the Author):

I appreciate the detailed and considered response to my query regarding the potential for maternal effects, and am satisfied by the response. I do wonder if some more of the information you provide in your response to reviewers (particularly the first two paragraphs) might be added to either the manuscript or supplementary information. This might alleviate concerns from readers who worry about the same issue, but is not essential.

Overall this is great work, and I very much look forward to seeing it in print.

Reviewer #2 (Remarks to the Author):

The authors have done a nice job of addressing comments, and the manuscript reads well. I still think this leaves a lot up in the air, as common garden experiments could either confirm or deny the opposite patterns reported in other studies. But that's another manuscript unto itself, and I think this is an important and solid study as-is.

I have two minor comments:

1) I missed this in the first read, but it's notable that the mean difference in metabolic rate between drainages is of similar magnitude to the difference between high/low predation populations within drainages. Given the magnitude of this effect, I wonder if it's worth a sentence or two to discuss this, or to direct the reader to a relevant section of supplementary material. Clearly, there's a lot of naturally occurring variation in metabolic rate not explained by body size, given that the two drainages don't appear to differ in mean body size.

I agree with the first reviewer's criticism about SMR possibly reflecting active growth, and with the authors' response. But, I don't see that addressed in the manuscript itself. Perhaps around L83 would be an appropriate place?

Reviewer #3 (Remarks to the Author):

I reviewed this manuscript before. My sole major concern was that authors use size at maturity as an index of pace-of-life. Authors responded by citing an earlier paper demonstrating that size at maturity is genetically positively correlated with age at maturity. This is not sufficient. A convincing demonstration of correlated evolution of metabolism and pace-of-life must constitute evidence for shifts in suites of life-history traits indicative of shifts in pace-of-life. As stated in my previous review, the demonstration of correlated evolution of metabolism and size at maturity is convincing and interesting but as such not a

demonstration of correlated evolution between metabolism and pace-of-life. In lieu of more convincing information, the general conclusions drawn here about the evolution of pace-of-life therefore do not follow from the data presented.

As a minor comment (line 278-279), authors should use bivariate mixed-effects models to appropriately calculate the covariance between metabolism and pace-of-life, which would enable the authors to control for appropriate fixed effects while avoiding inflated p-values caused by the anticonservative approach used now.

Reviewers' comments:

Reviewer #1 (Remarks to the Author):

Comment: I appreciate the detailed and considered response to my query regarding the potential for maternal effects, and am satisfied by the response. I do wonder if some more of the information you provide in your response to reviewers (particularly the first two paragraphs) might be added to either the manuscript or supplementary information. This might alleviate concerns from readers who worry about the same issue, but is not essential.

Overall this is great work, and I very much look forward to seeing it in print.

Reply: We agree that other readers may also have the same concern. For this reason, we have now: 1) integrated the first paragraph of our previous response into the methods section to provide rationale for our use of F1 generation fish (see lines 163-177), and 2) integrated the second paragraph of our previous response into the discussion section to acknowledge the potential role of maternal effects (see lines 111-113). Note that male size at maturity is no longer part of our analyses.

Reviewer #2 (Remarks to the Author):

Comment: The authors have done a nice job of addressing comments, and the manuscript reads well. I still think this leaves a lot up in the air, as common garden experiments could either confirm or deny the opposite patterns reported in other studies. But that's another manuscript unto itself, and I think this is an important and solid study as-is.

I have two minor comments:

Comment 1) I missed this in the first read, but it's notable that the mean difference in metabolic rate between drainages is of similar magnitude to the difference between high/low predation populations within drainages. Given the magnitude of this effect, I wonder if it's worth a sentence or two to discuss this, or to direct the reader to a relevant section of supplementary material. Clearly, there's a lot of naturally occurring variation in metabolic rate not explained by body size, given that the two drainages don't appear to differ in mean body size.

Reply: The reviewer is correct that there is a significant drainage effect that is independent of the predation effect. We do not yet have an explanation for this drainage effect but can at least cite the literature showing that the guppies from these two drainages are genetically quite different from one another^{1,2}, which is part of the basis for arguing that each represents an independent replicate in which guppies have adapted to life with and without predators. We can also show that there are parallel drainage effects on the pace of life of guppies from these rivers (see new analyses of SMR versus the overall pace of the entire life history (Fig. 2). We have now included a brief discussion of this drainage effect in the discussion section (see lines 113-115) and also provided a more detailed explanation of the evolutionary history of these populations as supplementary text.

Comment 2) I agree with the first reviewer's criticism about SMR possibly reflecting active growth, and with the authors' response. But, I don't see that addressed in the manuscript itself. Perhaps around L83 would be an appropriate place?

Reply: We have now added this into the statistical methods section since it provides rationale for why juveniles and adults were analysed together (lines 278-283).

Reviewer #3 (Remarks to the Author):

Comment: I reviewed this manuscript before. My sole major concern was that authors use size at

maturity as an index of pace-of-life. Authors responded by citing an earlier paper demonstrating that size at maturity is genetically positively correlated with age at maturity. This is not sufficient. A convincing demonstration of correlated evolution of metabolism and pace-of-life must constitute evidence for shifts in suites of life-history traits indicative of shifts in pace-of-life. As stated in my previous review, the demonstration of correlated evolution of metabolism and size at maturity is convincing and interesting but as such not a demonstration of correlated evolution between metabolism and pace-of-life. In lieu of more convincing information, the general conclusions drawn here about the evolution of pace-of-life therefore do not follow from the data presented.

Reply: At issue here is whether our study provides sufficient evidence that metabolic rate evolves alongside the life history (or pace of life). Previous transplant experiments and population comparisons under common garden laboratory conditions demonstrate that guppies evolve a slower paced life history when invading low predation environments¹⁻⁷. These evolutionary changes in the life history are repeated across multiple different lineages and involve changes in a whole suite of traits including older age and larger size at maturity in males, older age and larger size at first parturition in females, reduced reproductive rates (longer inter-litter intervals), and reduced reproductive investment (reproductive allotment - % dry weight devoted to reproduction). This body of research is well cited and has made Trinidadian guppies a textbook example of life history adaptation in the wild. As such, the novelty of our work here is not to demonstrate that life history evolution occurs in guppies, but rather to build on this large body of research and examine whether these changes in the life history are accompanied by parallel changes in energy metabolism. We do this by measuring the metabolic rates of guppy populations that are already known to exhibit differences in their pace of life (the one exception being Intro2008 population for which data are not yet available). We find that metabolic rates differ between guppy populations with a slow versus fast-paced life history and that these differences are consistent across multiple evolutionary transitions (both naturally occurring and experimentally manipulated) from a faster to slower-paced life history. To our knowledge, we also provide the first estimate for the rate of evolution of energy metabolism in the wild.

To more convincingly demonstrate the novelty of our study, we have now made the following changes to the manuscript:

1) We omitted our consideration of male size at maturity. It is not needed to demonstrate life history evolution in these populations since that has already been established by previous research (see above). We included it in our previous analyses because it is an accurate index of the evolution of the life history as a whole and because it allowed us to consider the Intro2008 population (for which there is no other life history data currently available) in our examination of how metabolic rates change with the life history across multiple populations. However, its inclusion only seems to create confusion over the novelty of our study.

2) We replaced our consideration of male size at maturity with an analysis of the relationship between metabolic rate and the overall pace of the life history using data on a whole suite of life history traits (male age and size at maturity, female age and size at first parturition, inter-litter interval, and reproductive allotment) collected in previous studies of these same populations. These life history data are from population comparisons (low versus high predation) conducted at separate times but using standardized protocols and under similar environmental conditions in the laboratory (see new section in methods on lines 233-257 and Supplementary Table 2). This analysis extends our within-drainage analyses of metabolic differences between guppy populations with a slow versus fast paced life history. It allows us to consider whether a single linear function can describe the relationship between metabolic rate and the pace of the life history across all six populations and thus without respect to predation level or drainage of origin. Using these data, we find that a single

PCA describes the majority of variation in these life history traits with an axis that delineates slow to fast-paced life histories (see lines 297-314 of main text). We also find that patterns across drainages (and thus independent transitions from a fast to slow paced life history) mirror those found within drainages: A single linear function ($r = 0.93$, $P = 0.008$) describes the relationship between metabolic rate and the pace of the life history across the 6 study populations (see Fig. 3). Together, these analyses provide multiple lines of evidence that metabolic rate evolves in parallel with the pace of the life history.

Comment: As a minor comment (line 278-279), authors should use bivariate mixed-effects models to appropriately calculate the covariance between metabolism and pace-of-life, which would enable the authors to control for appropriate fixed effects while avoiding inflated p-values caused by the anticonservative approach used now.

Reply: The purpose of this analysis was to examine whether a single function could describe the relationship between SMR and male size at maturity without regard to the effects of predation or drainage. This analysis was conducted on the means for each population. This analysis has now been replaced by analysis of the correlation between SMR and the pace of the life history. Since there are only two variables (SMR and life history score) in this model, we feel that it is appropriate to use a simple bivariate correlation. We have now included a statement about the analysis being conducted on the means for each population since this was not clear in the previous version of the manuscript (see lines 310-311). We have also now removed the standard errors from Fig. 2 since these were what likely created confusion over the exact details of the analysis.

References:

- 1 Reznick, D. N. & Bryga, H. Life-history evolution in guppies (*Poecilia reticulata*: Poeciliidae). V. Genetic basis of parallelism in life histories. *American Naturalist* **147**, 339-359 (1996).
- 2 Reznick, D. N., Rodd, F. H. & Cardenas, M. Life-history evolution in guppies (*Poecilia reticulata*: Poeciliidae). IV. Parallelism in life-history phenotypes. *American Naturalist* **147**, 319-338. (1996).
- 3 Reznick, D. N. The impact of predation on life history evolution in Trinidadian guppies: the genetic components of observed life history differences. *Evolution* **36**, 1236-1250 (1982).
- 4 Reznick, D. N. & Endler, J. A. The impact of predation on life history evolution in Trinidadian guppies (*Poecilia reticulata*). *Evolution* **36**, 160-177 (1982).
- 5 Reznick, D. N. Life history evolution in guppies. 2. Repeatability of field observations and the effects of season on life histories. *Evolution* **43**, 1285-1297 (1989).
- 6 Reznick, D. A., Bryga, H. & Endler, J. A. Experimentally induced life-history evolution in a natural population. *Nature* **346**, 357-359 (1990).
- 7 Reznick, D. N., Shaw, F. H., Rodd, F. H. & Shaw, R. G. Evaluation of the rate of evolution in natural populations of guppies (*Poecilia reticulata*). *Science* **275**, 1934-1937 (1997).

REVIEWERS' COMMENTS:

Reviewer #1 (Remarks to the Author):

I have now reviewed both the revised manuscript, and the responses to the reviewer comments from the last round of review. In my view, the responses to the reviewers address their concerns appropriately, and for my part I have no further suggestions for revision. I remain excited by the work, I think that it should be published, and look forward to seeing it in print.

Reviewer #2 (Remarks to the Author):

The new analysis directly incorporating life history data is more straightforward, and I think has improved the ms. It seems to address the concerns of the most critical reviewer, though I can't speak for them. As a minor comment, the figure caption and the results should specify that the analysis uses PC1, and ought to include the % variance explained.

REVIEWERS' COMMENTS:

Reviewer #1 (Remarks to the Author):

I have now reviewed both the revised manuscript, and the responses to the reviewer comments from the last round of review. In my view, the responses to the reviewers address their concerns appropriately, and for my part I have no further suggestions for revision. I remain excited by the work, I think that it should be published, and look forward to seeing it in print.

Reply: Thank you for your time and effort in reviewing our manuscript.

Reviewer #2 (Remarks to the Author):

The new analysis directly incorporating life history data is more straightforward, and I think has improved the ms. It seems to address the concerns of the most critical reviewer, though I can't speak for them. As a minor comment, the figure caption and the results should specify that the analysis uses PC1, and ought to include the % variance explained.

Reply: Thank you for your time and effort in reviewing our manuscript. We have now included the following statement in the figure legend: "The first PCA axis had an eigenvalue of 3.70, explained 61% of the variation in life history traits, and differentiated populations with late maturity and low reproductive investment (low PCA scores) from those with early maturity and high reproductive investment (high PCA scores)".